# Evaluation of Toxic Effects Induced by Sub-Acute Exposure to Low Doses of α-Cypermethrin in Adult Male Rats

**DOI:** 10.3390/toxics10120717

**Published:** 2022-11-23

**Authors:** Vilena Kašuba, Blanka Tariba Lovaković, Ana Lucić Vrdoljak, Anja Katić, Nevenka Kopjar, Vedran Micek, Mirta Milić, Alica Pizent, Davor Želježić, Suzana Žunec

**Affiliations:** Institute for Medical Research and Occupational Health, 10000 Zagreb, Croatia

**Keywords:** cholinesterase, genotoxicity, oxidative damage, subacute exposure, synthetic pyrethroids

## Abstract

To contribute new information to the pyrethroid pesticide α-cypermethrin toxicity profile, we evaluated its effects after oral administration to Wistar rats at daily doses of 2.186, 0.015, 0.157, and 0.786 mg/kg bw for 28 days. Evaluations were performed using markers of oxidative stress, cholinesterase (ChE) activities, and levels of primary DNA damage in plasma/whole blood and liver, kidney, and brain tissue. Consecutive exposure to α-cypermethrin affected the kidney, liver, and brain weight of rats. A significant increase in concentration of the thiobarbituric acid reactive species was observed in the brain, accompanied by a significant increase in glutathione peroxidase (GPx) activity. An increase in GPx activity was also observed in the liver of all α-cypermethrin-treated groups, while GPx activity in the blood was significantly lower than in controls. A decrease in ChE activities was observed in the kidney and liver. Treatment with α-cypermethrin induced DNA damage in the studied cell types at almost all of the applied doses, indicating the highest susceptibility in the brain. The present study showed that, even at very low doses, exposure to α-cypermethrin exerts genotoxic effects and sets in motion the antioxidative mechanisms of cell defense, indicating the potential hazards posed by this insecticide.

## 1. Introduction

The living environment is increasingly burdened by the excessive use of pesticides producing adverse biological effects against both target and nontarget species. Pyrethroids are a class of insecticides chemically derived from natural substances commonly regarded as relatively safe [1]. Their use has been increasing for the past decade, accompanied by a decline in the use of organophosphates with established high acute toxicity for mammals [2]. However, in pyrethroid synthesis, emphasis was put on improving efficiency, which may compromise the safety margin. A class of synthetic pyrethroid used on a large scale is cypermethrin [3]. The compound α-cypermethrin consists of highly active isomers of cypermethrin and belongs to type II pyrethroids due to the presence of a cyano group at the α-carbon of the alcohol moiety [4,5]. Products containing α-cypermethrin are broad-spectrum insecticides registered for agricultural use as a foliar application on food and feed crops, but also for use on industrial, commercial, and residential sites to control insect pests such as ants, cockroaches, and fleas [6]. Its primary insecticidal activity is the disturbance of the nerve membrane of the insect by delaying the closure or inactivation of voltage-sensitive sodium channels. Affected by α-cypermethrin, sodium channels can remain open for seconds, compared to the normal period of a few milliseconds [7,8], resulting in more sodium ions crossing and depolarizing the neural membrane beyond the normal extent, finally leading to neurotoxicity [9,10]. Besides interacting with sodium channels, α-cypermethrin may also interfere with other receptors of the nervous system, e.g., calcium and chloride channels, as secondary target sites [11,12], and accumulate in the brain, exerting neurotoxicity in nontarget organisms [13]. The residues of α-cypermethrin are commonly detected in the environment and food, but also in human urine and breast milk [5,14]. Its widespread exposure within the general population raises concern regarding its toxicity even at environmental concentrations. Humans are indirectly affected mainly by consuming pesticide-contaminated food. Despite the fact that literature data indicates its low toxicity, α-cypermethrin bioaccumulation and persistence in mammalian tissues are significant from a toxicological point of view [15,16]. Although the primary target of pyrethroids is the nervous system, studies have reported that cypermethrin exposure may be linked to reproductive toxicity, hepatotoxicity, immunotoxicity, and genotoxicity [17,18,19,20,21]. Induction of intracellular reactive oxygen species (ROS) and oxidative stress have been proposed as important mechanisms for the toxicity of pesticides, including α-cypermethrin [22,23,24,25]. An increase of ROS may induce damage to lipids, proteins, and DNA, leading to lipid peroxidation, protein oxidation, and DNA damage [26]. It has been suggested that cypermethrin binds to DNA through reactive groups, leading to a destabilization and unwinding of the DNA [27]. Cypermethrin administration caused DNA fragmentation in leukocyte cell nuclei of Swiss albino mice [28].

Besides its action via oxidative stress, previous studies have reported that cypermethrin induces toxicity through inhibition of acetylcholinesterase (AChE) activity [29,30,31,32]. Cholinesterases (ChEs) are enzymes found in the central and peripheral nervous system, particularly in nervous tissue, muscle, erythrocytes, liver, and plasma [33]. They catalyze the hydrolysis of choline-based esters, several of which serve as neurotransmitters [34]. Acetylcholinesterase (AChE; E.C. 3.1.1.7) and butyrylcholinesterase (BChE; E.C. 3.1.1.8) represent important targets in the toxicity of many insecticides. The inhibition of AChE is a useful biomarker of organophosphate and carbamate pesticides poisoning [35], while serum BChE has been used as an indicator of hepatic, renal and thyroid diseases, as well as a marker for pesticide toxicity [36]. 

While the majority of the studies on the toxic effects of isomers of cypermethrin were conducted by using dosages that were either lethal or sub-lethal, reports on the implications following exposure to doses relevant to human settings are rare. This comprehensive multidisciplinary study aimed to assess the potential effects of exposure on the kidney, liver, and brain of adult male Wistar rats at levels relevant for real-life scenarios set out in the current EU legislation and generally considered not harmful to humans. Cypermethrin consists of eight stereo-isomers, and according to their mixtures and the amount of each isomer in the mixture, it can be further divided into substances with ISO common names among which beta-cypermethrin is a newer form [37]. Since α-cypermethrin is much more in use, we wanted to fill up some gaps in its toxicity assessment [37,38]. Following a 28 day repeated dose exposure, we assessed physiological, biochemical, and genotoxic endpoints, including body and organ weight change, parameters of oxidant/antioxidant status, cholinesterase activity, and DNA damage, which enabled insight into the α-cypermethrin mode of action from different points of view. Considering the fact that females are generally more likely to incur DNA damage, we used only male rats [39]. Results of conducted study could contribute to better risk assessments for exposure to low doses of α-cypermethrin.

## 2. Materials and Methods

### 2.1. Experimental Compounds

The active ingredient α-cypermethrin (CAS-No. 67375-30-8), purity grade 100%, was purchased as the analytical standard Pestanal^®^ (Sigma-Aldrich Laborchemikalien GmbH, Seelze, Germany). To obtain a stock solution, the pesticide was dissolved in ethanol (EtOH), then diluted in deH2O to reach 0.03% EtOH. All other chemicals were of analytical grade and purchased from Sigma-Aldrich Chemical Co. (St. Louis, MO, USA), unless otherwise specified.

### 2.2. Animals and Experimental Design

The study included adult three-month-old male Wistar rats from an inbred colony at the Institute for Medical Research and Occupational Health (IMROH) in Zagreb weighing 340.1 ± 19.3 g and receiving standard rat pellet feed [GLP certified food Mucedola 4RF21 (Mucedola, Milan, Italy)] and water *ad libitum*. They were bred and housed under pathogen-free conditions in a steady-state microenvironment in accordance with the Guide for the Care and Use of Laboratory Animals [40]; air temperature 22 ± 2 °C, relative humidity 40–70%, and a 12/12 light/dark cycle. Animals were treated according to internationally accepted animal welfare guidelines. The study was approved by the Ethics Committee of IMROH, Zagreb, Croatia, and the Croatian Ministry of Agriculture (Reg no. 100-21/14-5, Class 01-18/14-02-2/6 of 11 June 2014).

Animals were divided into seven experimental groups (5 animals/group): three control groups and four treated groups. The negative control was kept in identical conditions, but received water instead of pesticide treatment. Positive control was treated with ethylmethanesulphonate (EMS), a well-established genotoxicant recommended for in vivo comet assay in rodents, at the dose of 300 mg/kg bw/day [41] over last three days. Solvent control received EtOH in saline, 0.03% solution. The first treated group was exposed to 2.186 µg/kg bw/day of α-cypermethrin, corresponding to Residential Exposure Level (REL) proposed by the European Food Safety Authority [42]. The second treated group was exposed to 0.015 mg/kg bw/day (dose corresponding to the Acceptable Daily Intake (ADI)) of α-cypermethrin. The third treated group was exposed to 0.157 mg/kg bw/day (dose corresponding to Occupational Exposure Limit (OEL)) of α-cypermethrin, and the fourth exposed group was exposed to five times the OEL (0.786 mg/kg bw/day).

At the start of the study, rats were weighed, and inspected by a licensed veterinarian at IMROH. During the experiment, rats were weighed weekly and the volume/concentration of applied α-cypermethrin solution was adjusted accordingly. Survival and clinical signs of poisoning were monitored on a daily basis. All groups (except positive control) were treated for 28 consecutive days by oral gavage, and received 1 mL of treatment solution once a day using a gastric probe. Animals were sacrificed 24 h after the last received dose. The body weight of rats was determined and compared with the initial body weight.

### 2.3. Collection of Blood and Tissues

At the end of the experiment, all animals were humanely euthanized by exsanguination under intraperitoneal Xylapan/Narketan anesthesia (Xylapan, Vetoquinol UK Ltd., Towcester, UK, 12 mg/kg bw *i. p.*/Narketan, Vetoquinol UK Ltd., 80 mg/kg bw). A licensed veterinarian at IMROH examined the animals for gross pathological changes of the internal organs. The blood samples were collected in heparinized vacutainers by a dissection of the carotid artery under general anesthesia and further processed. One portion of whole blood was immediately used to prepare agarose microgels for the alkaline comet assay while the portion used for biochemical analyses was centrifuged (980× *g*, 10 min, at +4 °C) to separate red blood cells and plasma that were stored at −20 °C until further processing. Livers, kidneys, and brains were dissected out and weighted. Based on the obtained values, relative liver, kidney, and brain weight were calculated using the following formula:Relative organ weight (ROW) = [absolute organ weight/body weight at sacrifice day] × 100.

Tissues were rinsed in cold PBS buffer (without Ca^2+^ and Mg^2+^), weighed, and divided into two portions intended for biochemical analyses and the comet assay. The samples used for biochemical analyses were immediately frozen in liquid nitrogen and stored at −80 °C until further processing while the tissue samples used for comet assay were prepared as described later in the text.

### 2.4. Biochemical Analysis

Before the measurements, tissue samples were homogenized (100 mg tissue/mL 50 mmol/L phosphate buffer, pH 7.8). Prior to the measurement of oxidative stress, parameters samples were centrifuged at 20,000× *g* and 4 °C for 30 min to obtain a supernatant. Before the measurements of ChE activities in brain tissues, the 100 mg/mL homogenates were diluted with 50 mmol/L potassium phosphate buffer containing 1 mmol/L EDTA, pH 7.4. 

Superoxide dismutase (SOD) activity in plasma and tissue supernatants was measured spectrophotometrically in accordance with the method by Flohé and Ötting [43]. The reduction rate of cytochrome c by superoxide radicals was monitored at 550 nm on a Cecil 9000 Spectrophotometer (Cecil Instruments Limited, Cambridge, UK) utilizing the xantine-xantine oxidase system as the source for O^2−^. SOD competed for superoxide and decreased the reduction rate of cytochrome c. One unit of SOD was defined as the amount of enzyme that inhibits the rate of cytochrome c reduction by 50%. Enzyme activity was expressed as IU/g protein.

Catalase (CAT) activity in plasma and tissue supernatants was determined by measuring the decrease in absorbance at 240 nm on a Cecil 9000 Spectrophotometer (Cecil Instruments Limited, Cambridge, UK) of a reaction mixture consisting of H_2_O_2_ (10 mmol/L final concentration) in phosphate buffer (pH 7.0) and the required sample volume [44]. Enzyme activity was calculated using the molar extinction coefficient (40 mmol/L/cm) and expressed as IU/g protein.

Glutathione peroxidase (GPx) activity in whole blood and brain tissue supernatants was determined spectrophotometrically according to the method described by Belsten and Wright [45]. The amount of GSH oxidized by t-butyl hydroperoxide was determined by following the decrease in the β-NADPH concentration, and the decrease in absorbance at 340 nm was measured spectrophotometrically. One unit of GPx is the number of micromoles of β-NADPH oxidized per minute. The results were expressed as IU/g hemoglobin (Hb) (whole blood GPx) or IU/g protein (tissue supernatant GPx).

The end products of lipid peroxidation, e.g., malondialdehyde (MDA), were measured using thiobarbituric acid reactive substances (TBARS) assay with some modification by Drury et al. [46]. Butylated hydroxytoluene (BHT; 5 µL, 0.2% *w*/*v*) and phosphoric acid (750 µL, 1% *v*/*v*) were added to 50 µL of plasma or tissue homogenate samples and mixed. Afterwards, 250 µL 0.6% (*w*/*w*) thiobarbituric acid (TBA) and 445 µL H_2_O were added, and the reaction mixture was incubated in a water bath at 90 °C for 30 min. The mixture was cooled in an ice bath and absorbance was measured at 532 nm on a Shimadzu UV Probe Spectrophotometer (Kyoto, Japan). The TBARS concentration was calculated using standard curves of increasing 1,1,3,3-tetramethoxypropane concentrations, and expressed as µmol/L.

Total cholinesterase (ChE), acetylcholinesterase (AChE), and butyrylcholinesterase (BChE) activities were determined in the plasma and tissue homogenate samples using spectrophotometric Ellman method [47]. Enzyme activity was measured in a 0.1 mol/L sodium phosphate buffer, pH 7.4, at 25 °C using ATCh (1.0 mmol/L) and DTNB (0.3 mmol/L). AChE and BChE activities were distinguished using the BChE-selective inhibitor ethopropazine (20 µmol/L). Increase in absorbance was monitored at 412 nm over 4 min. All of the measurements were performed on a Cecil 9000 Spectrophotometer (Cecil Instruments Limited, Cambridge, UK). Enzyme activity was expressed as IU/g protein. 

Protein content was measured by the Bradford assay [48] using bovine serum albumin as the standard while Hb in whole blood was measured spectrophotometrically at 540 nm (Cary 50 UV/Vis, Varian Inc., Palo Alto, CA, USA) by the standard cyanmethemoglobin method using Hemiglobincyanide standard (Mallinckrodt Baker B.V., Denver, Holland, Dublin, Ireland).

### 2.5. DNA Damage Detected by the Alkaline Comet Assay

The comet assay was carried out using the procedure described by Singh et al. [49] with slight modifications for blood and various tissues (blood [50], liver and kidney [51], and brain [52]). Preparation of tissues for the alkaline comet assay was done within one hour following animal sacrifice. Livers and kidneys were dissected and rinsed in cold TBS buffer [50 mmol/L Tris-Cl, 150 mmol/L NaCl, pH 7.5] until as much blood as possible was removed. 

To release single cells from dissected tissues, a small piece of tissue was put in a chilly mincing buffer [75 mmol/L NaCl (Kemika, Zagreb, Croatia) and 24 mmol/L Na_2_EDTA, pH 7.5] and minced with a pair of fine scissors. The obtained cell suspensions were kept on ice for a few seconds to allow large clumps to settle, and immediately used to prepare agarose microgels for the alkaline comet assay. Two replicate slides were prepared for each animal and experimental point, using whole blood or cell suspensions.

We used “sandwich” agarose microgels, made of four agarose layers. Their first layer consisted of a 1% normal melting point (NMP), which was used for pre-coating of slides. The second gel layer was 0.6% NMP agarose. The third layer consisted of a 0.5% low melting point (LMP) agarose mixed with 10 μL of heparinized whole blood (10 μL per slide) or 10 μL of liver, kidney, or brain cell suspension per slide. Finally, 0.5% LMP agarose was applied as the top layer over the gel-embedded cells. After solidification of the gel on an ice-cold metal tray, the slides were submerged in freshly prepared ice-cold lysis solution (100 mmol/L EDTA, 2.5 mol/L NaCl, 10 mmol/L Tris-Cl, pH 10, 1% of Triton-X 100 and 10% DMSO) at 4 °C overnight (except brain tissue, for at least 1 h). The slides were quickly washed with distilled water (to remove residues of detergents and salts), placed in a vertical Coplin jar in ice-cold freshly made alkaline electrophoresis buffer (1 mmol/L Na_2_EDTA and 300 mmol/L NaOH, pH > 13), and kept in the dark for DNA unwinding (blood and brain: 20 min at 4 °C; liver and kidney: 10 min at 4 °C). Then, slides were transferred into a horizontal electrophoresis unit (Horizon 11.14, Whatman, Maidstone, UK), filled with electrophoresis buffer. Electrophoresis was carried out in the same buffer with different conditions applied for the different tissues: blood: 0.86 V/cm, 300 mA, 20 min at 4 °C; liver and kidney: 1 V/cm, 300 mA, 10 min at 4 °C; brain: 1 V/cm, 300 mA, 20 min at 4 °C.

After electrophoresis, slides were rinsed in Tris buffer, pH 7.5 (3 × 5 min). After a final wash, they were dehydrated in 70% and 96% EtOH (10 min in each), air dried, and stored in boxes in dry conditions. Prior to the comet measurements, slides were stained with 20 μg/mL ethidium bromide. Microscopic analyses were performed on a fluorescence microscope (Olympus BX51, Olympus, Tokyo, Japan) (under 200× magnification) using the Comet Assay IVTM image analysis software (Instem-Perceptive instruments Ltd., Suffolk, Halstead, UK). 

As the main descriptor of DNA damage in this study, we used the percentage of DNA in the tail (or tail intensity). We measured a total of 100 comets randomly captured on each slide at the constant depth of the gel, avoiding the edges of the gel and air bubbles [53]. Nucleoids with >80% DNA in the tail region were excluded from the analysis, as according to literature data, they may represent DNA damage resulting from cytotoxicity [54,55]. 

In liver samples, medium-sized cells (parenchymal cells or hepatocytes, between 30 and 40 μm of head length) and small-sized cells (nonparenchymal cells, <30 μm of head length) were recorded separately [56]. 

The reported results are based on the measurements of 1000 comets per each experimental group and the cell type (i.e., 2 slides per an individual and cell type (= 200 comets) × 5 individuals per group). To avoid the variability, one well-trained scorer scored all of the comets.

### 2.6. Statistical Analysis

The statistical analysis was performed using Dell™ Statistica™ licensed statistical software package Version 13.5.0.17 (TIBCO Software Inc., Palo Alto, CA, USA). Normality of distribution was tested with the Kolmogorov–Smirnov test. The results for body and organ weight were expressed as mean ± standard error. The results for biochemical markers and comet assay were expressed as median and range.

Normally, distributed data (body and organ weight) were analyzed by repeated measures analysis of variance (one-way ANOVA), followed by Tukey’s multiple comparison test. Since most of the results for biochemical markers were not normally distributed, we used the Kruskal–Wallis test followed by the Median test for between group comparisons. The same was applied for the analysis of the results obtained by comet assay. Values were considered statistically significant at *p* < 0.05.

## 3. Results

Treatment with α-cypermethrin for 28 days did not cause any signs of systemic toxicity or death in adult male Wistar rats. In the course of study, the exposed rats did not produce marked behavioral changes compared to control rats or show signs typical for cholinergic overstimulation such as hypothermia, tremors, incoordination, lacrimation, diarrhea, and salivation. Gross necropsy did not reveal any treatment-related findings.

### 3.1. Body and Organ Weight

At the end of the treatment, significantly lower body weights were observed in animals treated with 0.015, 0.157, and 0.786 mg/kg bw/day of α-cypermethrin compared to negative control (*p* = 0.00003). In comparison to negative control, all of the groups treated with α-cypermethrin had significantly lower body weight gains (*p* < 0.00001). The differences in absolute liver weight (*p* < 0.00001), relative liver weight (*p* < 0.00001), and absolute kidney weight (*p* = 0.0047) of rats treated with 0.015 and 0.157 mg/kg bw/day were statistically significant compared to negative control, with lower values observed in α-cypermethrin-treated rats. No significant change was found in relative kidney weight and absolute brain weight. However, a significant increase in the relative brain weight of rats treated with 0.015 and 0.157 mg/kg bw/day compared to negative control was determined (*p* = 0.0356) (Table 1).

### 3.2. Biochemical Markers of Toxicity

Data for activities of antioxidant enzymes SOD, CAT, and GPx, as well as for the measured levels of lipid peroxidation expressed as concentration of TBARS are shown in Figure 1. The effect of α-cypermethrin on individual markers of oxidative stress differed by tissue. Continuous administration of low doses of α-cypermethrin for 28 consecutive days did not affect SOD activity in the liver and brain of the treated rats. Although the graphical representations for the measured SOD activities indicate different medians for the treated groups, this is due to a larger measured range of minimum and maximum values. In the kidney, treatment with the lowest (2.186 µg/kg bw/day) and the highest (0.786 mg/kg bw/day) dose resulted in a decrease of SOD activity, although this difference was significant only when the lowest dose was administered (*p* = 0.0043). Opposite to that, administration of 0.157 mg/kg bw/day resulted in the highest measured SOD activity although not statistically different from the negative control. In plasma, α-cypermethrin caused a slight decrease in SOD activity, especially after administration of higher doses (0.157 and 0.786 mg/kg bw/day), but these changes were not statistically significant compared to the negative control. In all tissues, the range of measured SOD activities in the solvent group was similar to the values in the negative control. There were no changes in CAT activity in the examined tissues and plasma after 28 day exposure to low doses of α-cypermethrin. GPx activity significantly increased in liver (*p* = 0.00012) and brain tissue of rats treated with α-cypermethrin (*p* = 0.00001). A comparison between groups showed that GPx activity was significantly increased in the livers of animals treated with doses corresponding to 0.015 and 0.157 mg/kg bw/day, compared to the negative control group. GPx activity was also significantly higher in rats treated with doses 2.186 µg/kg bw/day and 0.786 mg/kg bw/day in comparison to the group treated with 0.157 mg/kg bw/day. In brain tissue, a significant increase in GPx activity was observed in all of the applied doses of α-cypermethrin, compared to negative control (*p* < 0.0001). Exposure to α-cypermethrin had no effect on the activity of GPx in kidney. A significant decrease in GPx activity was observed in blood of rats treated with doses corresponding to 2.186 µg/kg bw/day, 0.015, and 0.157 mg/kg bw/day, compared to negative control (*p* < 0.0001). A significant difference was also observed in blood GPx activity between groups treated with doses corresponding to 0.015 and 0.786 mg/kg bw/day.

Treatment with α-cypermethrin did not cause changes of TBARS concentration in the liver, kidney, and plasma. The measured values were similar to the negative control. An increase in TBARS concentration was noticed in the brain tissue of treated rats, especially in animals treated with 2.186 µg/kg bw/day of α-cypermethrin (*p* = 0.0014).

Table 2 reports the results concerning effects of α-cypermethrin on the catalytic activity of cholinesterases in the rats’ liver, kidney, brain, and plasma. Total ChE and AChE activities were decreased in the liver (ChE, *p* = 0.0083; AchE, *p* = 0.0018) and kidney (ChE, *p* = 0.0003; AchE, *p* = 0.0004) of a α-cypermethrin-treated rats, although those changes were significantly different compared to control only in rats treated with doses 2.186 µg/kg bw/day and 0.786 mg/kg bw/day. Since in the rats’ brains AChE accounted for 90% of the total measured cholinesterase activity, we demonstrated only those results for brain tissue. However, α-cypermethrin exposure did not affect AChE activity in the brain of the exposed rats. Concerning BChE, a fluctuation of activities was noticed. In the liver of α-cypermethrin treated rats, a decrease in BChE activity was detected, being significant compared to control only for treatment with 0.015 mg of α-cypermethrin/kg bw/day (*p* = 0.0013). In the kidney of α-cypermethrin treated rats, BChE activities were similar to the control. In plasma of α-cypermethrin treated rats, administration of doses 2.186 µg/kg bw/day, 0.015 or 0.157 mg/kg bw/day resulted in decreased BChE activities, while administration of 0.786 mg/kg bw/day increased BChE activity resulting in a significant difference between these groups (*p* = 0.0071).

### 3.3. The Alkaline Comet Assay

Figure 2 shows the results of the alkaline comet assay expressed as comet tail intensity (%DNA) measured in peripheral blood leukocytes, brain, kidney, and liver cells (parenchymal and nonparenchymal) of male Wistar rats after a 28 day oral exposure to α-cypermethrin. 

In peripheral blood leukocytes, tail intensity was significantly lower than in positive control in all of the exposed animals (*p* < 0.0001). No significant difference was observed between the α-cypermethrin-treated groups and negative controls. A between-groups comparison showed that tail intensity was significantly higher in blood cells of rats treated with 0.015 mg/kg bw/day compared to those treated with 0.786 mg/kg bw/day of α-cypermethrin. 

Tail intensity measured in brain cells of rats treated with α-cypermethrin was significantly lower than in positive control, and significantly higher than in negative and solvent control at all of the applied doses (*p* < 0.0001). In brain cells of rats treated with the lowest dose of α-cypermethrin (2.186 µg/kg bw/day), tail intensity was significantly lower compared to the other treated groups. Additionally, in brain cells of rats treated with 0.015 mg/kg bw/day, tail intensity was significantly lower compared to the group treated with 0.786 mg/kg bw/day of α-cypermethrin, which points to a dose-related increase of DNA damage.

In kidney cells, tail intensity was significantly lower in all groups exposed to α-cypermethrin in comparison to positive and solvent control (*p* < 0.0001). In rats treated with 2.186 µg/kg bw/day, 0.015 mg/kg bw/day, and 0.157 mg/kg bw/day, significantly lower DNA damage was observed in comparison to negative controls. Tail intensity measured in kidney cells of rats treated with the highest dose of α-cypermethrin (0.786 mg/kg bw/day) was significantly higher than in rats treated with doses 2.186 µg/kg bw/day and 0.157 mg/kg bw/day.

In small or nonparenchymal liver cells, tail intensity was significantly lower than in positive controls in all of the exposed animals (*p* < 0.0001). A significantly higher tail intensity was measured in cells of rats treated with 0.015 mg/kg bw/day or 0.157 mg/kg bw/day in comparison to negative control and doses 2.186 µg/kg bw/day and 0.786 mg/kg bw/day. Similarly to nonparenchymal cells, in parenchymal (medium) liver cells or hepatocytes, tail intensity was significantly lower than in positive controls in all of the exposed animals (*p* < 0.0001). A significantly higher tail intensity was measured in cells of rats treated with 0.015 mg/kg bw/day compared to negative and solvent control and all of the other groups treated with α-cypermethrin. Tail intensity measured in parenchymal liver cells of animals treated with the highest dose of α-cypermethrin was significantly lower than in the solvent control group and rats treated with 0.157 mg/kg bw/day.

## 4. Discussion

The extensive use of pyrethroids for agricultural and domestic applications has led to adverse effects in many nontarget species and made the toxicity assessment of these compounds a public health imperative. This encouraged us to study the effects that may occur after sub-chronic oral exposure at the concentration levels of α-cypermethrin potentially encountered in everyday life. The applied concentrations were calculated based on several toxicological reference values: residential exposure level, acceptable daily intake, and occupational exposure limit. Considering that diet is the main route of human exposure to pyrethroids, α-cypermethrin was applied orally to adult male rats. We analyzed blood and tissue biomarkers as the endpoints that can be disrupted after ingestion of α-cypermethrin and found that sub-acute exposure to very low doses of α-cypermethrin may impair the antioxidative defense system, decrease the ChE and AChE activities in liver and kidney and lead to DNA instability in liver, kidney, and brain of exposed rats.

### 4.1. Effect on Body and Organ Weight

Body and organ weight analysis represents an important endpoint in the assessment of pesticide toxicity [57]. Such information contributes to an accurate explanation of the data obtained using other methods, for instance genotoxicity assays. In the present study, consecutive 28 day exposure to α-cypermethrin resulted in significantly less body weight gain in all of the treated groups. Lower kidney weight, liver weight and relative liver weight, and higher relative brain weight of rats was also observed, compared to control group. A significant reduction in animal weight gain despite unlimited access to food is a clear indication of the general toxicity of this insecticide. The significant changes in body weight gain were confirmed in related rat studies where cypermethrin was administered orally at much higher doses. The treatment of adult male Sprague Dawley rats with 13.15, 18.93, and 39.66 mg of cypermethrin/kg bw/day for 12 weeks, resulted in a significantly lower animal weight gain [58], and the same was observed after a 15 day oral exposure to 12.5, 25, and 50 mg of α-cypermethrin/kg bw/day [59]. 

The significantly lower kidney and liver weight was in accordance with the reduced weight gain of rats treated with α-cypermethrin, however, the results obtained for organ weights differ between studies. A significant reduction in liver and kidney weights was found in mice treated for 28 days with 250 mg of cypermethrin/kg bw/day, compared to the control group. According to the authors, the main reason for the lower weight of liver and kidney in treated mice was the damage caused by its accumulation in such high concentrations and the excessive work of these organs to excrete cypermethrin from the body [28]. The liver has an important role in the decomposition and removal of toxic compounds such as pesticides, while kidneys are responsible for the elimination of metabolic waste from the body [4]. In a study by Mansour et al., a 28 day oral administration of α-cypermethrin to female rats at a dose of 0.05 mg/kg bw/day resulted in significantly lower weights of the liver and kidney compared to the control group [60]. Histological examination of liver sections showed hemorrhagic areas in the dilated hepatic sinusoids and active Kupffer cells, while in kidney sections, intraglomerular hemorrhagic areas and degeneration in the renal tubules was observed [60]. On the other hand, although a downward trend of weights of liver and kidney was observed in rats treated with 12.5–50 mg of α-cypermethrin/kg bw/day, no significant differences were noted between the treated groups and the control group [59]. 

In addition to the effects on liver and kidney weight, the results of our study showed that rats treated with α-cypermethrin had significantly higher relative brain weight in comparison to negative controls. The neurotoxicity of cypermethrin has been evaluated in acute, sub-chronic, and developmental neurotoxicity studies, showing inconsistent findings regarding changes in brain weight. Our results are in accordance with the ones obtained by Mansour et al. [60] where the absolute and relative weights of brain were significantly higher in rats treated with 0.05 mg of α-cypermethrin/kg bw/day. A decrease in brain weight was observed in rats exposed daily to sub chronic levels of cypermethrin in a four-week experiment by Kalra and Sangha [61]. Both female and male rats treated with cypermethrin showed extensive neuronal damage and marked neuronal degeneration and the severity of the effect increased with the increasing dose of cypermethrin [61]. A nonsignificant decrease was also observed in brain weight of rats exposed to 3.83 mg of cypermethrin/kg bw for 7 days [62]. Due to its lipophilic nature, cypermethrin can easily accumulate in the brain and take part in the pathogenesis of various neurological disorders [9], which is the main concern regarding exposure to pyrethroids.

### 4.2. Cholinesterase Activity

Cholinesterases are recognized with special concern in the toxicology of all pesticides affecting the nervous system, as tissue inhibition of ChEs may be useful as a surrogate marker. The repeated 28 day exposure of rats to α-cypermethrin in our study changed the ChE, AChE, and BChE activity in kidney and liver, but did not significantly affect enzyme activities in plasma and brain tissue. The differential distribution of different molecular forms of ChE as well as their mode of interaction with pyrethroids could be a possible reason for observed patterns of ChE inhibition in different organs. For instance, a marked decrease in the AChE levels was previously observed in rat livers with cirrhosis. On the other hand, liver BChE, whose decrease in human serum has been associated with chronic liver disease, was apparently unaffected [63]. Besides its role in inactivation of blood-circulating acetylcholine, AChE in hepatocytes may be involved in some inter- and intracellular regulatory mechanisms, such as regulation of cell growth and cell adhesion [34,63,64]. Exposure to sub-lethal concentrations of cypermethrin for a period of 30 days inhibited the AChE activity in different organ tissues of the fish, including liver and kidney [65], while treatment with β-cypermethrin decreased the ChE activity in liver and kidney of frogs *Euphlyctis cyanophlyctis* [66]. The inhibition of AChE by cypermethrin may be a consequence of the binding of this insecticide in the enzyme active or peripheral site or both. More precisely, cypermethrin may interact with the anionic substrate binding site [29], and due to its high lipophilicity, also bind at the hydrophobic surface of AChE [67]. Regarding the ChE activities in plasma and in the brain, previous studies have reported that exposure to sub-chronic doses of cypermethrin decreases the levels of AChE in a dose-dependent manner in the plasma and brain of female and male rats [61]. A 7 day treatment with 3.83 mg of cypermethrin/kg bw also decreased the activity of AChE in the brain of male rats [62]. In the present study, the rats repeatedly received α-cypermethrin in doses that were possibly too low to exert a clear inhibitory effect on plasma and brain ChE, in comparison to previous reports when much higher doses were applied [29,61,62]. Furthermore, there is no fixed relationship between peripheral cholinesterase inhibition and brain acetylcholinesterase inhibition, which varies for individual agents. It is important to know the nature of reaction of pyrethroids with ChE (reversible or irreversible inhibition) as well as with tissue concentration and tissue half-life. Another explanation for such results could lie in the adaptive processes during chronic and sub-chronic exposures to low level doses of pesticides. Brain AChE appears to be unrelated to the liver, kidney, and erythrocyte type, concerning de novo synthesis and turnover [68]. 

### 4.3. Oxidative Stress Response

During normal cellular energy production and metabolism in mitochondria, ROS are constantly formed in low concentrations, taking part in many physiological actions in the cell metabolism, cell growth, development, and differentiation. However, in conditions of increased levels of these species, damage to cell functions may occur, as they can harm cellular proteins, DNA, and lipids [69,70]. Since ROS have extremely short half-lives and are difficult to measure directly, the TBARS assay, which relies on the ability of secondary products of lipid peroxidation and other reactive aldehydes to react with TBA, is often used for lipid peroxidation estimation [71,72]. The enzymatic antioxidant defense system plays a critical role in protecting cells from the surplus of formed reactive species. Enzymes including SOD, CAT, GPx, glutathione reductase (GR), and glutathione S-transferase (GST) are the first line of defense against free radicals, acting together to reduce the toxic effects of the ROS: SOD converts superoxide to hydrogen peroxide (H_2_O_2_), which is then converted in water by CAT and GPx [73]. 

The generation of ROS and oxidative stress have been described as the most important mechanisms by which pyrethroids exert their cellular action. As a lipophilic molecule, cypermethrin can easily pass through the cell lipid bilayer and damage its integrity [74]. The results obtained for TBARS levels in the brain tissue of the treated rats confirm that sub-chronic oral exposure to low doses of α-cypermethrin produces significant amounts of oxidative damage. This was accompanied by the increase in the activity of GPx in brain tissue as well as in the liver of all treated groups. Due to the large amount of polyunsaturated fatty acids and relatively low antioxidant system, the brain is highly vulnerable to oxidative injury [29], which may explain the increase in brain TBARS levels. The histopathological examination of brain and liver tissues and oxidative damage biomarkers showed adverse effects of α-cypermethrin in rats treated with the dose corresponding to 1/10 LD_50_ for 28 days [75]. The treatment of rats with 3.83 mg of cypermethrin/kg bw for 7 days resulted in an elevation in lipid peroxidation, a decrease of glutathione (GSH) and total protein content, and inhibition of antioxidant enzymes activity in the brain [62]. The observed increase in enzyme activity in the brain and liver of rats treated with α-cypermethrin may be explained as a compensatory adaptive response to the increased presence of free radicals, intending to minimize the toxic effects of the ROS generated by the pyrethroid. A similar increase in activities of GST, GPx, and GR in rat brains was reported by Kalra and Sangha [61] after treatment with cypermethrin at doses corresponding to 1/25 and 1/50 LD_50_ values. Contrary to our results, several recent studies report a decrease in antioxidant enzyme activity in the liver of cypermethrin-treated rats [19,75,76]. However, it must be noted that in the mentioned studies, doses similar to LD_50_ values were tested, which may explain the observed differences. α-cypermethrin-induced oxidative damage in liver and kidney was also reported in other organisms, such as lizards [4] and fish [77].

Unlike the results obtained for the brain and liver, the blood GPx activity of animals treated with α-cypermethrin was significantly inhibited. A decrease in the activity of GPx is usually accompanied by the depletion of GSH, which is used as a substrate by this enzyme in conditions of excessive ROS formation [78]. One of the limitations of our study is that we did not measure the concentration of GSH in blood/tissues of rats. However, such findings have been reported in other studies on pyrethroid-exposed animals [16,26]. The mechanism of inhibition of the antioxidant enzymes may also include the inhibition of transcription of specific genes, resulting in decreased mRNA levels of certain enzymes and reflecting in their lower activities. In a study by Soliman et al. [74], cypermethrin decreased both the activity and mRNA expression of the analyzed antioxidants in rats following oral administration of 12 mg/kg bw for 28 days. The absence of the effect of α-cypermethrin on the measured parameters in other organs can be attributed to the very low doses applied to the present study. Together with differences in the duration and/or routes of exposure employed, this is presumably the main reason for the discrepancies observed between studies. 

### 4.4. Genotoxic Effects of Exposure to α-Cypermethrin

The comet assay or the single-cell gel electrophoresis assay is a well-established direct method to detect a broad spectrum of DNA damage. It enables the detection of DNA single and double strand breaks, alkali-labile sites, oxidatively damaged DNA bases, apurinic and apyrimidinic sites, and DNA-DNA and DNA-protein crosslinks [79]. The comet assay or the single-cell gel electrophoresis assay also predicts the genotoxicity of potentially mutagenic or carcinogenic substances [80]. With regard to other genotoxicity tests, such as micronucleus assay, chromosomal aberrations and sister chromatid exchanges, the comet assay has sufficient sensitivity for detecting low levels of DNA damage, and has been applied in the evaluation of the possible genotoxicity of different pesticides [81,82,83].

In the present study, values of tail intensity measured in the studied cell types of exposed rats, were significantly different from negative control at almost all of the applied doses. When interpreting the results of the comet assay, we should consider that they reflect the damage levels detected at the final point of the experiment, which lasted for 28 days. During that period, due to the constant delivery of the tested compound, cells are faced with the induction of many types of DNA lesions which they try to counteract via various repair mechanisms. Therefore, the net damage measured using the comet assay represents an array of the damage levels recorded in the single cells, pointing both to the cell sensitivity and the efficiency of the repair. In line with the results obtained for oxidative stress, the highest degree of DNA damage was observed in the brain of α-cypermethrin-treated rats, confirming that the increase in lipid peroxidation followed by an activation of antioxidant enzymes, primarily GPx, plays a major role in cypermethrin-induced genotoxicity. Increased levels of 8-hydroxy-2′-deoxyguanosine, which is one of the predominant forms of free radical-induced oxidative stress in nuclear and mitochondrial DNA, have been measured in rat liver and urine following 28 day oral exposure to cypermethrin [76,84]. It has also been reported that cypermethrin may cause DNA damage in rats’ brains, as evidenced by an increase in the percentage of DNA damage level, percentage of DNA in tail, and tail moment at doses representing 1/10 and 1/30 LD_50_ applied for 28 days [85]. Similarly, Hammad and Ziada [75] reported DNA damage in the brain and liver of male albino rats indicating α-cypermethrin-induced genotoxicity after oral exposure to dose corresponding to 1/10 LD_50_. Hepatic DNA damage after exposure to sub-lethal doses of cypermethrin has been observed in several other in vivo studies [26,76,86,87]. Đikić et al. [86] reported that cypermethrin damaged DNA at alkali-labile sites, and that the observed genotoxic alterations in hepatocytes demonstrated its clastogenic properties. Small hydrophobic molecules, such as cypermethrin, can easily pass through the cell membrane and reach the cell nucleus. Once within the nucleus, cypermethrin may bind to DNA through the reactive groups of its acid moiety and form DNA monoadducts and DNA inter strand crosslinks [27,88]. Moreover, vinyl and dimethylcyclopropane groups, which are an integral part of the cypermethrin structure, can be oxidized. The formed active metabolites (methyl butanol and vinyl) may induce DNA damage, leading to destabilization and unwinding of the DNA [89]. However, the data regarding genotoxicity after exposure to low doses of cypermethrin are lacking. The genotoxic action of cypermethrin has recently been demonstrated in the offspring of females exposed to doses considered safe (0.05 mg/kg/day) [90]. Cypermethrin induced a significant increase in the micronucleus frequencies and comet assay parameters in the blood and liver cells of the dams as well as in the blood, liver, and brain cells of the pups. As the authors suggest, genetic damage caused by cypermethrin in dams and in the offspring, may be the result of an increase in ROS levels, which directly affects the DNA molecule causing breaks and mutations [87,90]. Considering that, in the present study, changes in oxidative stress response were observed in rat brains and livers, we can presume that both direct and indirect toxic effects were responsible for the genome instability as measured by the alkaline comet assay. Although a dose-dependent hepatotoxicity was observed in previous experiments [21,87], this was not the case in our study. The expected half-life of double-strand breaks repair is slower in low-dose exposure than in cases of higher/high dose exposure [91], which may explain the absence of a dose–response relationship in the present study. Additionally, endocrine disruptive chemicals and pesticides such as pyrethroids are a classic example of compounds with a nonmonotonic dose response effect, exerting severe effects even at low doses while the same severity may not be anticipated at higher doses [10,92]. Our results suggest that kidney cells expressed lower levels of primary DNA damage than the other studied cell types. This could be explained by a possibly lower uptake of the tested chemical and/or their reactive metabolites, capable of producing measurable DNA instability. Another explanation could be that processing of agarose microgels itself contributes to the loss of the most damaged cells, which thus escape or avoid measurement, and the final value obtained for the tail intensity descriptor refers only to the nucleoids with lower levels of DNA damage. Nevertheless, cypermethrin toxicity at the level of kidney in rats must be further evaluated, considering that Liu et al. [93] recently showed that β-cypermethrin has nephrotoxic characteristics. Exposure of male rats to β-cypermethrin abnormally altered renal histomorphology and ultra-structures, induced renal DNA damage, and impaired renal functions. Exposure to β- cypermethrin activated the JNK/c-Jun pathway by inducing ROS and oxidative stress. Taken together, data obtained using the alkaline comet assay indicate that, at the applied doses, cypermethrin caused genomic instability in different rat tissues, which obviously depended on the inherited susceptibility of each cell type and its capacity to repair DNA lesions inflicted during the period of 28 day repeated exposure.

## 5. Conclusions

To the best of our knowledge, this is the most comprehensive study that deals with the physiological, biochemical, and genotoxic effects of very low doses of α-cypermethrin as a whole. The applied doses are considered safe by regulatory agencies as they represent the residential exposure level, acceptable daily intake, and occupational exposure limit. Still, the current study indicates that sub-acute exposure of rats to such low doses of α-cypermethrin may result in oxidative stress responses, inhibition of ChE activity, and low-level DNA instability. Although the study confirms that the primary target of pyrethroids is the nervous system, it is evident that they can be toxic to other organ systems.

Our inability to include histological analyses may be considered the limitation of the study. However, the fact that such low doses have the potential to produce measurable biological effects points to their importance and calls for more extensive research. Chronic exposure to pesticides in humans is already an issue of great concern for both the general population and agricultural workers, and an understanding of the interaction they have with an organism is critical. Moreover, pesticides are rarely applied as single compounds, and there is a huge lack in knowledge on their effects in mixtures regarding cumulative and synergistic effects. Future studies should therefore consider aggregate exposure, as well as the exposure to the same substance from different independent sources and/or via different pathways. Such studies will be of potential benefit to public health and to policy makers to mitigate the dangers provoked by pesticides.

## Figures and Tables

**Figure 1 toxics-10-00717-f001:**
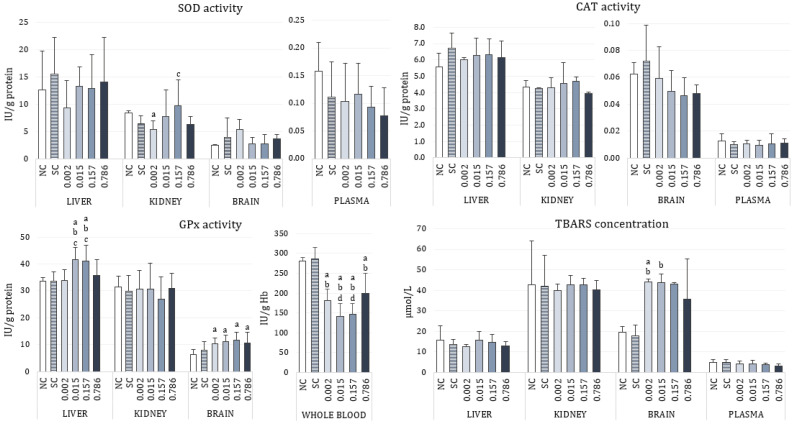
Changes of oxidative stress parameters in the liver, kidney, brain, and plasma/whole blood of adult male Wistar rats orally exposed to different doses (mg/kg bw/day) of α-cypermethrin for 28 consecutive days. The results are shown as median (box) and quartile range (whisker) values. SOD—superoxide dismutase; CAT—catalase; GPx—glutathione-peroxidase; TBARS—thiobarbituric reactive substances. NC—negative control, SC—solvent control. Statistical significance (Kruskal–Wallis test followed by Median test; One-way ANOVA followed by Tukey’s test for GPx activity in kidney and whole blood) was set at *p* < 0.05: ^a^—vs. negative control, ^b^—vs. solvent control, ^c^—vs. dose 2.186 µg/kg bw/day, ^d^—vs. dose 0.786 mg/kg bw/day.

**Figure 2 toxics-10-00717-f002:**
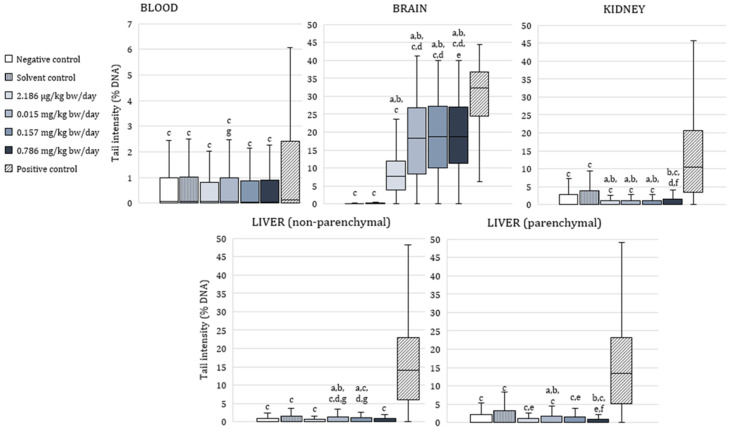
Results of the alkaline comet assay expressed as tail intensity (%DNA) in peripheral blood leukocytes, brain, kidney, and liver cells of adult male Wistar rats (N = 5 rats per group) treated orally for 28 consecutive days with α-cypermethrin and in the respective controls. Data are presented as median (line), 25th and 75th percentile (box), and range (whisker). Statistical significance was set at *p* < 0.05 (Kruskal–Wallis test followed by Median test; ^a^—vs. negative control, ^b^—vs. solvent control, ^c^—vs. positive control, ^d^—vs. dose 2.186 µg/kg bw/day, ^e^—vs. dose 0.015 mg/kg bw/day, ^f^—vs. dose 0.157 mg/kg bw/day and ^g^—vs. dose 0.786 mg/kg bw/day).

**Table 1 toxics-10-00717-t001:** Body, liver, kidney, brain, and relative organ weights of adult male Wistar rats in control and α-cypermethrin-treated groups.

Parameters	Negative Control	Positive Control	Solvent Control	2.186 µg/kg bw/Day	0.015 mg/kg bw/Day	0.157 mg/kg bw/Day	0.786 mg/kg bw/Day
Initial weight/g	333 ± 2	312 ± 3	307 ± 3	357 ± 8	337 ± 3	360 ± 6	342 ± 7
Final weight/g	409 ± 8 ^c^	341 ± 6 ^a,b,d^	388 ± 6 ^c,e^	392 ± 8 ^c,e^	346 ± 7 ^a,b,d^	369 ± 9 ^a^	366 ± 9 ^a^
Weight change/g	76.6 ± 9.5 ^c^	29.0 ± 5.8	80.4 ± 7.00 ^c,d,e,f,g^	35.6 ± 2.6 ^a,b,e,f^	9.40 ± 4.58 ^a,b,d^	8.80 ± 4.99 ^a,b,d^	24.0 ± 2.81 ^a,b^
Liver weight/g	12.4 ± 0.5 ^c,e,f^	9.24 ± 0.18 ^a,b,d^	11.41 ± 0.34 ^c,e,f^	11.70 ± 0.47 ^c,e,f^	8.39 ± 0.34 ^a,b,d,g^	8.68 ± 0.45 ^a,b,d,g^	10.96 ± 0.41 ^e,f^
Relative liver weight	3.03 ± 0.09 ^e,f^	2.71 ± 0.04 ^f^	2.94 ± 0.09 ^e,f^	2.98 ± 0.07 ^e,f^	2.42 ± 0.06 ^a,b,d,g^	2.35 ± 0.07 ^a,b,c,d,g^	2.99 ± 0.08 ^e,f^
Kidney weight/g	1.29 ± 0.05 ^a,e,f^	1.10 ± 0.03 ^a^	1.15 ± 0.03	1.22 ± 0.03 ^e^	1.01 ± 0.02 ^a,d^	1.06 ± 0.04 ^a^	1.16 ± 0.05
Relative kidney weight	0.31 ± 0.01	0.32 ± 0.01	0.30 ± 0.01	0.31 ± 0.01	0.29 ± 0.01	0.29 ± 0.01	0.31 ± 0.01
Brain weight/g	1.38 ± 0.05	1.33 ± 0.04	1.44 ± 0.04	1.52 ± 0.03	1.41 ± 0.04	1.44 ± 0.05	1.47 ± 0.06
Relative brain weight	0.34 ± 0.02 ^e,g^	0.39 ± 0.01	0.37 ± 0.01	0.39 ± 0.002	0.41 ± 0.01 ^a^	0.39 ± 0.01 ^a^	0.40 ± 0.01

Values are given as mean ± S.E. Statistical significance was set at *p* < 0.05; ^a^—vs. negative control; ^b^—vs. solvent control; ^c^—vs. positive control; ^d^—vs. dose 2.186 µg/kg bw/day; ^e^—vs. dose 0.015 mg/kg bw/day; ^f^—vs. dose 0.157 mg/kg bw/day; ^g^—vs. dose 0.786 mg/kg bw/day (One-way ANOVA followed by Tukey’s test).

**Table 2 toxics-10-00717-t002:** Changes in cholinesterase activities (IU/g protein) in liver, kidney, brain, and plasma of adult male Wistar rats orally exposed to different doses (mg/kg bw/day) of α-cypermethrin for 28 consecutive days.

	Liver			Kidney			Brain	Plasma		
	ChE	AChE	BChE	ChE	AChE	BChE	AChE	ChE	AChE	BChE
**Negative control**	0.467 0.412–0.549	0.215 0.170–0.304	0.241 0.227–0.274	0.204 0.163–0.227	0.124 0.095–0.148	0.084 0.039–0.112	23.9 20.0–35.9	0.079 0.077–0.108	0.063 0.056–0.084	0.023 0.014–0.030
**Solvent** **control**	0.244 ^a^ 0.212–0.274	0.169 0.138–0.188	0.074 ^a^ 0.056–0.125	0.262 0.166–0.282	0.180 0.122–0.219	0.062 0.039–0.090	22.5 13.6–31.5	0.046 ^c^ 0.042–0.089	0.030 ^a^ 0.024–0.054	0.019 0.010–0.035
**2.186** **µg/kg bw/day**	0.245 ^a^ 0.207–0.276	0.101 ^a^ 0.073–0.153	0.134 0.092–0.167	0.096 ^a,b^ 0.084–0.111	0.079 0.058–0.106	0.019 0.0–0.049	25.6 22.3–34.7	0.068 0.044–0.089	0.047 0.032–0.060	0.016 0.012–0.032
**0.015** **mg/kg bw/day**	0.284 0.212–0.413	0.155 0.139–0.264	0.129 ^a^ 0.073–0.149	0.136 0.102–0.188	0.066 ^b^ 0.048–0.083	0.088 0.031–0.105	25.4 20.0–30.9	0.067 0.043–0.081	0.043 0.031–0.072	0.012 ^c^ 0.007–0.029
**0.157** **mg/kg bw/day**	0.259 0.235–0.333	0.125 0.098–0.182	0.137 0.131–0.159	0.128 0.101–0.142	0.064 ^b^ 0.052–0.074	0.069 0.037–0.087	27.1 25.6–33.9	0.053 0.051–0.081	0.048 0.042–0.056	0.008 ^c^ 0.0–0.027
**0.786** **mg/kg bw/day**	0.247 ^a^ 0.216–0.264	0.094 ^a^ 0.072–0.122	0.151 0.132–0.158	0.098 ^a,b^ 0.090–0.101	0.052 ^a,b^ 0.047–0.067	0.038 0.032–0.053	29.5 26.0–32.2	0.117 0.087–0.142	0.059 0.047–0.073	0.059 0.034–0.069

ChE—total cholinesterases; AChE—acetylcholinesterase; BChE—butyrylcholinesterase. The results are presented as median and range (minimum and maximum values). Statistical significance (Kruskal–Wallis test followed by Median test) was set at *p* < 0.05: ^a^—vs. negative control, ^b^—vs. solvent control, ^c^—vs. 0.786 mg/kg bw/day.

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
