# Peer review of "Evaluation of Toxic Effects Induced by Sub-Acute Exposure to Low Doses of α-Cypermethrin in Adult Male Rats"

_toxics, 2022, doi:10.3390/toxics10120717_

Round 1

Reviewer 1 Report

Dear authors,

The manuscript was well written and quite comprehensive, covering all the major aspects of toxicity endpoints related to cypermethrin. There are a few changes that need clarification.

Line 31-33 - The statement is good, but are there any recent studies showing increase in the usage of α-cypermethrin and decline of organophosphate pesticides.

Line 38-39 - Same comment as above, use a citation

Line 74 - Regarding Cholinesterases, are you specifically talking about AChE? If including BChE, It was not mentioned about plasma. 

Line 1-92 - The first 2 paragraphs is too long. The flow is missing through the introduction, overall introduction is very descriptive, it will be better to make it more concise and straight to the point. Please include citations.

Line 124 & 126 - "weighted" this means you made them look heavy or did you check the weight of the rats weekly? If you check the weight then correct this to "weighed".

Line 262 - In results sections please include the p values in text. It will be helpful to follow the significant results.

In figures it is too busy and sometimes confusing to follow NC or a. It will be better to use two letters or one letter all over the figure for consistency.

In discussion, you are repeating the same points from introduction. For example functions of AchE. It is better to cut down the discussion part and focus on the positive results.

Author Response

Response to the Reviewer #1 of the manuscript ID: toxics-2028631, entitled: “Evaluation of toxic effects induced by sub-acute exposure to low doses of α-cypermethrin in adult male rats” (by Vilena Kašuba, Blanka Tariba Lovaković, Ana Lucić Vrdoljak, Anja Katić, Nevenka Kopjar, Vedran Micek, Mirta Milić, Alica Pizent, Davor Želježić and Suzana Žunec).

The Authors would like to thank the Reviewer #1 for her/his valuable comments regarding our manuscript. We have responded to all of the comments and revised the paper accordingly. Detailed responses to each comment are provided in the attached Word document

Reviewer 2 Report

In this study presented by Kasuba et al., effects of low dose alpha-cypermethrin on male rats were investigated. Overall, this study was well designed and the data was clearly presented. It fits the topic of MDPI Toxics and can be accepted with minor revision.

I just have a minor comment in the Introduction part, the authors are encouraged to mention about the rationale of choosing alpha-, rather than beta-cypermethrin, and only investigated in male rats. 

Otherwise, this is a solid study and the discussion was sufficient.

Author Response

Response to the Reviewer #2 of the manuscript ID: toxics-2028631, entitled: “Evaluation of toxic effects induced by sub-acute exposure to low doses of α-cypermethrin in adult male rats” (by Vilena Kašuba, Blanka Tariba Lovaković, Ana Lucić Vrdoljak, Anja Katić, Nevenka Kopjar, Vedran Micek, Mirta Milić, Alica Pizent, Davor Želježić and Suzana Žunec).

The Authors would like to thank the Reviewer #2 for her/his valuable comments regarding our manuscript. We have responded to all of the comments and revised the paper accordingly. Detailed responses to each comment are provided in the attached Word document
